# The Moderating Effect of Physical Activity on the Relationship between Sleep and Emotional Distress and the Difference between Blacks and Whites: A Secondary Data Analysis Using the National Health Interview Survey from 2005–2015

**DOI:** 10.3390/ijerph18041718

**Published:** 2021-02-10

**Authors:** Jesse Moore, Shannique Richards, Collin Popp, Laronda Hollimon, Marvin Reid, Girardin Jean-Louis, Azizi A. Seixas

**Affiliations:** 1Department of Population Health, New York University Langone Health, New York, NY 10016, USA; shannique.richards@nyulangone.org (S.R.); collin.popp@nyulangone.org (C.P.); laronda.hollimon@nyulangone.org (L.H.); girardin.jean-louis@nyulangone.org (G.J.-L.); 2Department of Community Medicine & Psychiatry, University of the West Indies, Kingston JMCJS2, Jamaica; marvin.reid@uwimona.edu

**Keywords:** emotional distress, mental health, physical activity, sleep, minority health, race

## Abstract

(1) Background: Unhealthy sleep durations (short and long sleep) are associated with emotional distress (ED). Minority populations, specifically Blacks, are more burdened with unhealthy sleep durations and ED. The ameliorative effect of physical activity (PA) on ED and sleep duration may provide insight into how to reduce the burden among Blacks and other minorities. However, it is unclear whether PA attenuates the relationship between sleep and ED, and whether this relationship differs by race. (2) Methods: We analyzed data from the nationally representative 2005–2015 National Health Interview Survey (NHIS) dataset. ED, physical activity, and sleep duration were collected through self-reports. Regression analyses investigated the moderating effect of PA on the relationship between sleep and ED (adjusting for age, sex, BMI, and employment status) and stratified by race. (3) Results: We found that sleep duration was independently associated with ED. Physical activity moderated the relationship between sleep and ED, the full population, and Whites, but not Blacks. (4) Conclusion: PA moderated the relationship between short, average, or long sleep and ED, but in stratified analyses, this was only evident for Whites, suggesting Blacks received differing protective effects from physical activity. Further research should be performed to understand the connection of physical activity to sleep and mental health.

## 1. Introduction

### 1.1. Emotional Distress and Race

Emotional distress, which is generally described in the literature as feelings of anxiety and depression [1], is a serious public health burden in the United States, as approximately 15% of the U.S. population has anxiety and/or depressive symptoms [2]. Although emotional distress is prevalent across the United States, epidemiological data indicate that Blacks are the most burdened. Acute and chronic daily and systemic stressors have been attributed to this emotional distress burden and have been further linked to worse mental health outcomes among Blacks compared to other racial/ethnic groups [3]. This heavy psychological burden among Blacks [3,4] has insidious effects on other health outcomes, such as cardiovascular disease and all-cause mortality. Although the roots of this burden are multifarious, partly due to a confluence of biological, behavioral, clinical, environmental, and psychosocial factors, focusing on the proximal behavioral risk factors may present the most immediate and impactful strategy to understand and address the burden of emotional distress and mental illness among Blacks.

### 1.2. The Link between Emotional Distress and Sleep Disparities among Blacks

Seminal work demonstrates that the high prevalence of poor sleep health outcomes (short duration, poor quality, sleep disorders, disrupted sleep timing, and circadian rhythms) among Blacks is linked with the disproportionate burden of emotional distress compared to other racial/ethnic groups. Overwhelming evidence shows a clear bidirectional association between insufficient/short sleep duration (≤6 h per 24 h period) and emotional distress, as sleep deprivation inhibits emotion and stress regulation, while acute and chronic levels of stress have shown to disrupt sleep patterns, leading to disturbed sleep [5,6]. However, more recent evidence highlights that both short (≤6 h) and long (≥9 h) sleep durations are independently associated with emotional distress and mental illness and may be responsible for racial/ethnic vulnerabilities in these populations [6,7]. The burden of emotional distress and unhealthy sleep (short or long sleep durations) among Blacks can be plausibly explained by various factors, such as the increased allostatic load due to institutional racism [6], a high prevalence of sedentary behavior [6], and disparities in healthcare, which are all associated with poor sleep and mental health literacy [7,8,9,10]. Mechanistically, poor and dysregulated sleep can cause functional deficits in the prefrontal cortex and amygdala in the brain, which heightens responses to negative stimuli and decreased moods [11,12]. The literature also posits that a full cycle of sleep “resets” the regulatory system in the prefrontal–amygdala circuit and increases the emotion and stimulus regulation [13]. However, studies show that Blacks may not benefit from the reset effects of sleep, as their sleep architecture differs from other racial groups. Blacks generally experience less time in deep sleep (stage N3 characterized by slow-wave sleep, which is linked to biological and brain restoration) than Whites, which may compromise their ability to maintain optimal mental health and cope with emotional distress [12,14]. Previous research found that REM sleep is particularly effective at resolving emotional distress [13]; therefore, a lack of efficiency in deep sleep, REM sleep, and higher sleep deprivation may provide a biological explanation of why sleep and emotional distress are particularly pronounced and linked in Blacks.

### 1.3. Physical Activity as a Moderator between Emotional Distress and Sleep

Physical activity may moderate the association between sleep and emotional distress by regulating arousal processes to affect sleep onset and quality. It has been independently linked with emotional distress/mental health outcomes and sleep duration. Given physical activity’s close relationship to both emotional distress and sleep, it may provide novel insights on the burden of and link between sleep deprivation and emotional stress among Blacks. In epidemiological studies, Blacks report less physical activity than the rest of the population, where only 39% meet the Centers for Disease Control and Prevention’s (CDC’s) recommended levels for physical activity, and 25% are sedentary [15]. Moreover, Blacks appear to be more susceptible to sleep-related (sleep deprivation, poor sleep quality, and sleep disorders) anxiety and depressive symptoms [16]. Studies show that moderate-to-vigorous physical activity interventions protect against emotional distress and increases sleep duration and the percentage of REM sleep [17,18].

### 1.4. Gap in the Literature

Despite the foregoing evidence linking the higher burden of poor sleep outcomes with emotional distress (especially among Blacks) and the ameliorative effects of physical activity on emotional distress and poor sleep and mental health outcomes, there is a relative dearth of research examining how physical activity may affect the relationship between sleep and emotional distress among Blacks relative to Whites. To fill this gap, our current study investigated whether (1) associations between unhealthy sleep durations, emotional distress, and physical activity, and (2) the moderating effect of physical activity on the unhealthy sleep duration and emotional distress association differed between Black and White adults in a large U.S. population-based sample

## 2. Materials and Methods

### 2.1. Participants and Dataset

The study used data from the 2005–2015 National Health Interview Survey (NHIS), which is a nationally representative dataset containing information extracted by trained interviewers in a cross-sectional sample from across the United States. The NHIS is part of the CDC Integrated Public Use Microdata Series, which is a library of datasets that are available for public and research use. Participants in the NHIS were administered questionnaires to obtain demographic, socioeconomic, and medical data. The total sample of the NHIS dataset included 416,152 participants. As part of the demographic interview, participants were asked to self-identify their race. This study focused on the racial/ethnic difference between Black and White participants; those who identified as other races/ethnicities (Asian, Hispanic, etc.) were excluded from the analysis, resulting in a final sample of 274,041 participants (46,218 Blacks (16.9%) and 227,823 Whites (83.1%)).

### 2.2. Procedure

Data were collected on a yearly basis for the duration of the sample taken for the study (10 years total) by interviewers trained by the U.S. Census Bureau. Households were also identified by the U.S. Census Bureau, and data was collected from all regions of the United States. There were no restrictions on who the data was collected from, provided participants were over the age of 18. The U.S. Census Bureau chose addresses to interview for these surveys to generate nationally representative data. The sampling selections for these households were as follows: first, all counties in the United States were divided into 15 groups based on their characteristics and one county was selected from each group. Next, within each county, smaller groups (with many households) were formed, and then 20 to 24 of these groups were selected. Finally, all the houses or apartments of that small group were identified and a sample of about 30 households was selected from within each group. This procedure is explained in more detail in the NHIS dataset Supplementary Material [19]. If a household had limited English fluency, an interpreter or other language assistance programs may have been used.

### 2.3. Measures

Sleep Duration: To ascertain the total sleep duration, participants were asked, “How many hours of sleep do you get on average in a 24 h period?” This information was coded as a categorical variable: if participants reported 7–8 h of sleep a night, they were coded as “healthy” sleep; if they got 6 or fewer hours, they were coded as “short” sleep; if they got 9 or more hours, they were coded as “long” sleep. In a separate variable, “healthy sleep” was coded as a binary variable with “unhealthy” sleep, which was considered all values that were not 7–8 h of sleep. Hours of sleep coded in this way were always rounded to the nearest hour.

Emotional Distress: Emotional distress is defined as feelings of anxiety and depressed mood, as well as symptoms of a nonspecific negative affect [1]. Emotional distress was measured using the Kessler-6, which is a validated six-item screening scale that asks participants to rate their frequency of feeling sad, nervous, restless, hopeless, worthless, and burdened over the last 30 days [1]. Responses are based on a five-point Likert-type scale, ranging from 1 (“not at all”) to 5 (“all of the time”). The total Likert-type scores, in a range from 6–30, were converted to the Kessler-6 scale from 0 to 24. Higher scores indicate greater levels of emotional distress symptoms. The present study treated emotional distress as a dichotomous variable, using the Kessler-6 guidelines to indicate an emotionally distressed status, as defined by Kessler. If a participant scored greater than 13 on the Kessler-6 survey, they were coded as having significant emotional distress; if a participant scored 13 or less, they were coded as not having significant emotional distress according to the Kessler-6 cutoff.

Physical Activity: Participants were asked, “How much physical activity (in hours, and rounded to the nearest hour) do you perform per day?” To capture the intensity of physical activity, participants were asked to respond to both moderate and vigorous activity separately. They were also asked to respond to the duration of their moderate and/or vigorous physical activity, measured in minutes, and reported the frequency of physical activity per week. For our study, we added the duration for both moderate and vigorous physical activity to create a dichotomous variable describing whether they regularly performed physical activity with either moderate and/or vigorous intensity.

Covariates: Our analyses adjusted for the confounding effects of age, sex, employment status, and body mass index (BMI), which are all associated with sleep duration, physical activity, and emotional distress [5]. In particular, the literature shows that employment status is particularly important when viewing sleep and mood in the context of race [20]. Age was self-reported and coded as a continuous variable, BMI was calculated by dividing the participants’ body weight in kilograms by their height in meters squared (treated as a continuous variables), sex was coded as a dichotomous variable (0 = male and 1 = female), and employment status was coded in terms of whether an individual was employed or unemployed.

### 2.4. Data Analysis

Multiple logistic regression models were performed to analyze the moderating effect of physical activity on the relationship between sleep (predictor) and emotional distress (outcome). Descriptive statistics were run to set up the analysis (Table 1). Two sets of models were performed. The first investigated the moderating effect of physical activity (PA) on the relationship between an unhealthy sleep duration and ED in Blacks and Whites. The second set of models investigated the moderating effect of PA on the relationship between short sleep and emotional distress (ED) and long sleep and ED in Blacks and Whites. All models fully adjusted for the confounding effects of age, sex, BMI, and employment were included. Bivariate correlations between all variables of interest as well as covariates in the full and stratified White or Black population are included. All analyses used the SPSS statistical program (SPSS version 25, IBM, Armonk, NY, USA).

## 3. Results

### 3.1. Descriptive Statistics

In Table 1, 45.2% (*n* = 123,805) of the participants identified as male and 54.8% (*n* = 150,236) identified as female. A total of 83.1% of the participants in the sample were White (*n* = 227,823) and 16.9% were Black (*n* = 46,218) (other races were not included). The mean age was 48.04 years and ranged from 18 to 85 years old. The average amount of sleep per night was 7.15 h; this number was slightly lower in Blacks (7.05 h) than in the full population and in Whites (7.17 h). A total of 67.1% of the White sample was physically active, and 57.3% of the Black sample was physically active. The average Kessler-6 emotional distress score was 8.50. The average Kessler score was 8.60 among Blacks and 8.48 among Whites. In the full sample, 12.3% reached the threshold to be considered emotionally distressed. Black participants had a slightly higher prevalence of emotional distress (13.8%), while 12.0% of Whites were emotionally distressed. Before the regression models were analyzed, correlations were explored to gauge the univariate relationships between the variables of interest and the covariates. The findings revealed that all the observed correlations were relatively weak but statistically significant, confirming the appropriateness of their inclusion in the regression models. All variables of interest were significantly correlated in the full model and in the race-stratified models. Further descriptive statistics (Table 2, Table 3 and Table 4).

### 3.2. Inferential Statistics

Logistic regressions were performed to determine the associations between the three variables of interest (sleep, emotional distress, and physical activity) to provide support for a full moderation model. Regressions were performed in the full model first; then, these regressions were stratified by race. In the full model, there was found to be an association between unhealthy sleep and emotional distress (odds ratio (OR) = 2.58, 95% confidence interval (CI) = 2.52–2.64, *p* < 0.001), physical activity and sleep (OR = 0.80, 95% CI = 0.78–0.81, *p* < 0.001), and physical activity and emotional distress (OR = 0.62, 95% CI = 0.61–0.64, *p* < 0.001). These relationships resulted in outcome likelihoods of participants being 2.58 times more likely to be emotionally distressed if getting unhealthy sleep, 20% less likely to get healthy sleep if not performing physical activity, and 38% less likely to be emotionally distressed if performing physical activity, respectively; sleep predicted emotional distress much more strongly than the associations in the other relationships. In the model that was stratified to include only Whites, there was found to be an association in sleep and emotional distress (OR = 2.64, 95% CI = 2.47–2.71, *p* < 0.001), physical activity and sleep (OR = 0.77, 95% CI = 0.76–0.79, *p* < 0.001), and emotional distress and physical activity (OR = 0.60, 95% CI = 0.59–0.62, *p* < 0.001). In the model stratified to include only Blacks, sleep and emotional distress (OR = 2.29, 95% CI = 2.16–2.42, *p* < 0.001) and emotional distress and physical activity (OR = 0.74, 95% CI = 0.71–0.79, *p* < 0.001) were found to be associated. Notably, physical activity was not found to predict healthy sleep in Blacks (*p* = 0.17). These relationships resulted in outcome likelihoods of White participants being 1.62 times more likely to be emotionally distressed if getting unhealthy sleep, 22.9% more likely to get unhealthy sleep if not performing physical activity, 40% more likely to be emotionally distressed if not performing physical activity and Black participants being 1.64 times more likely to be emotionally distressed if getting unhealthy sleep and 26.4% more likely to be emotionally distressed if not performing physical activity, demonstrating that most of these variables are associated and also that their associations differ between races.

A regression analysis model was constructed to test the hypothesis that physical activity moderates the relationship between sleep and emotional distress. This model included main effects for sleep and physical activity, along with the sleep and physical activity interaction effect. As seen in Table 2, there was a significant correlation between sleep and emotional distress (*r* = −0.10, *p* < 0.001). In the logistic regression, if the full population including covariates (age, sex, BMI, and employment), individuals with unhealthy sleep were 25.45% more likely of reporting Kessler-6 diagnosed emotional distress (OR = 2.55, 95% CI = 2.48–2.66, *p* < 0.001). Physical activity was also significant in the model, showing that individuals who did not perform at least some amount of regular physical activity had a 22% higher chance to report emotional distress (OR = 0.78, 95% CI = 0.73–0.85, *p* < 0.001). The interaction effect of physical activity and sleep was also significant in the model, with *p* < 0.001; therefore, people who both perform physical activity and get healthy sleep had a 7.1% lower chance to report emotional distress (OR = 0.929, 95% CI = 0.87–0.98, *p* = 0.003).

### 3.3. Race-Stratified Analysis

In the fully adjusted model of White participants, unhealthy sleepers were 1.6 times more likely to report emotional distress (OR = 2.60, 95% CI = 2.50–2.71, *p* < 0.001) compared to healthy/average sleepers. Physical activity also had a significant effect in the model; those who reported performing regular physical activity had a 29% lower chance of reporting emotional distress (OR = 0.71, 95% CI = 0.68–0.73, *p* < 0.001). The interaction effect achieved significance as well, meaning that White people who both performed physical activity and got healthy sleep had a 7% lower chance of reporting emotional distress (OR = 0.93, 95% CI = 0.88–0.98, *p* = 0.004).

In the fully adjusted model of Black participants, unhealthy sleepers had more than a two-fold likelihood of reporting emotional distress (OR = 2.42, 95% CI = 2.24–2.63, *p* < 0.001). Physical activity also had a significant effect in the model, as those who did not performed at least some physical activity were 15% more likely to report emotional distress (OR = 0.85, 95% CI = 0.78–0.93, *p* < 0.001), which was a weaker connection than for Whites. Notably, the interaction effect was not significant in the model for Black participants at *p* = 0.072.

Finally, supplemental analyses investigating the moderating effect of PA on the relationships between short sleep and ED and long sleep and ED (Table 5, Table 6, Table 7, Table 8, Table 9, Table 10, Table 11, Table 12 and Table 13) indicated that short sleepers had greater odds of ED relative to long and average/healthy sleepers. White short and long sleepers had a greater likelihood of ED compared to their Black counterparts. The attenuating effect of physical activity on the relationship between sleep duration and ED was greater among long sleepers relative to short sleepers and among Whites compared to Blacks.

## 4. Discussion

There were several important findings from this study. First, we found that sleep had a significant effect on emotional distress. This finding is consistent with previous research that has consistently demonstrated that unhealthy sleep durations (short or long sleep) are associated with emotional distress [21,22,23]. We also found an association between physical activity and sleep and between physical activity and emotional distress, and demonstrated that these associations also occurred when the population was stratified across Black and White participants. Second, we found that physical activity significantly moderated the relationship between sleep and emotional distress. This is also supported by the literature, which suggests that physical activity may be an important health behavior that influences both sleep and mental health [22,24,25]. Our findings were consistent with a biobehavioral model depicted in literature that suggests that emotional distress is heightened by a failure of a regulatory system in the amygdala and prefrontal cortex to “reset” itself in sleep and that unhealthy sleep deprives the brain of an opportunity to dissipate emotional distress [13]. The function of physical activity to promote REM sleep, which can also help to resolve emotional distress, was also reflected in our findings [13]. Stratified analyses by race (White and Black participants) indicated that this effect may differ by race/ethnicity, where the attenuating effect of physical activity on emotional distress was significant among Whites but not Blacks. This is consistent with the literature suggesting the vulnerability of Black populations to disparities in sleep, emotional distress, and sedentary behavior, and suggests that there may be other vulnerabilities amongst Blacks that attenuate the full benefit of exercise [5,16,26,27]. In addition, the increased allostatic load of stress on Blacks [28] may mean that regular healthy sleep and exercise may help to alleviate emotional distress, but it may not be fully effective to resolve the greater distress levels amongst Blacks.

### 4.1. Moderating Role of Physical Activity on the Sleep–Emotional Distress Association

The association between sleep and emotional distress was stronger for those who were physically active compared to those who were more sedentary. Our findings are supported by other research that has also found a stronger sleep and emotional distress link in physically active individuals. For example, a systematic review by Grgic et al. showed that a majority of physical activity studies found a stronger relationship in the link between regular physical activity and mental or physical health outcomes in active individuals than sedentary individuals [29]. However, the authors found that most studies did not incorporate sleep duration as a major focus, and the emotional distress literature involving the link between sleep and physical activity was lacking. This finding has precedence in the literature when using interaction models with logistic regression, especially with physical activity. Lovasi et al. stated that the odds ratio interaction effects may be misleading when analyzing variables that are already linked. The authors give an example from one of their previous studies examining age group and fast food restaurant proximity on obesity; they found a much larger odds ratio for the interaction effect than expected because less densely populated areas were less likely to have fast food restaurants, but were much more likely to be obese because very rural areas that are not densely populated have a high prevalence of obesity due to other factors, such as sedentary behavior [30]. Literature defines this effect in logistic regression as the “interaction fallacy” [31]. This phenomenon may help to explain our finding that people who exercise and have healthy sleep in the full population and the White sample were more likely to be emotionally distressed; people who sleep well are more likely to exercise regularly and may be more likely to report emotional distress because they may be more health literate and may be able to articulate their mental health status [9].

### 4.2. Racial/Ethnic Differences in the Moderating Role of Physical Activity on the Sleep–Emotional Distress Association

The analysis of the relationship between our three main variables of interest indicated differences across Black and White participants. Although significant effects of sleep and physical activity were found in all stratified regression models, the interaction effect between sleep and physical activity was not significant in the Black stratified sample. This finding that race differentially affects the moderation of physical activity between sleep and emotional distress is also supported in the literature. It is possible that emotional distress in Blacks is impacted by a larger range of external variables than for Whites (e.g., economic disadvantage, comorbid medical problems, stressful living conditions). For example, Pager and Shepherd stated that Blacks experience differential stressors in a broad range of domains of life in the United States, including employment, housing, banking, and other aspects of contemporary society [32]. Furthermore, the higher sleep needs of Blacks may affect this relationship: in addition to the need for more REM sleep in Blacks [14], they also have a shorter circadian rhythm than Whites, affecting the ability to recover from a night of poor sleep [33]. These findings, along with the increased amount of sedentary behavior in Blacks [15], may complicate the interaction between sleep and exercise on emotional distress. Future research would further examine why there do not seem to be the same significant relationships between health outcomes in Blacks as there are in Whites.

### 4.3. The Comparison in Short vs. Long Sleep in Predicting Emotional Distress

This study found that short sleep (≤6 h of sleep) predicted emotional distress more strongly than long sleep (9+ hours of sleep). Although the literature describes a “U-shaped” pattern of health outcomes, in that worse health outcomes are generally found when comparing either short or long sleep to healthy sleep, this relationship remains poorly understood. For example, when comparing short and long sleep to healthy sleep, studies found both metabolic and cardiovascular outcomes [34] and psychiatric and substance use disorders [35]. There is a dearth of research comparing the emotional distress outcomes among short or long sleepers. The literature posits the increased health risk from an irregular sleep schedule that comes from shift work as one of the underlying mechanisms for this relationship [19], but it remains unclear why short or long sleep may result in higher emotional distress scores.

### 4.4. Strengths and Limitations

There were several strengths of this study. The extremely large and nationally representative sample allowed this study to provide a reliable estimate of the relationship between variables that had been observed in smaller samples in previous research. Furthermore, the large number of variables collected in the data allowed for the inclusion of relevant covariates and stratification by race. In addition, emotional distress was measured using a widely used, though relatively simple, index of distress.

However, there are several limitations that affect the conclusions of this study. For example, because this is a secondary analysis of nationwide self-reported data, the variables of interest were not measured directly and are therefore subject to errors that come from self-reporting. Participants may have trouble accurately estimating their hours of sleep or levels of physical activity or may have different interpretations of these questions. For example, people could have had different interpretations of what is meant by “moderate” physical activity. Future studies could use validated physical activity questionnaires, such as the International Physical Activity Questionnaire. In addition, more thorough measures of emotional distress may identify aspects of distress that were not identified by the Kessler-6. Further research should seek to understand these associations in more precise measures and should seek to refine conceptualizations of these variables to accurately measure the qualities of physical activity on sleep and emotional distress.

## 5. Conclusions

This study examined the role of physical activity on the relationship between sleep and emotional distress among Blacks and Whites. The findings from our study demonstrated that Blacks did not benefit from the protective effects of physical activity as much as their White counterparts. Our study also demonstrated that both short and long sleep had a greater positive relationship with emotional distress than healthy sleep, but short sleep was the most positively associated. Further research should be performed to examine what qualities of physical activity affect this relationship, understand how sleep and physical activity differentially affect racial groups, and what other factors might drive this discrepancy.

## Figures and Tables

**Table 1 ijerph-18-01718-t001:** Group comparison of key variables between Blacks and Whites (*n* = 274,041).

Variable	White Population	Black Population	Chi-Square Comparison *p*-Value
Physically active	67.1%	57.3%	<0.001
Sleep duration			
≤6 h	58.7%	60.5%	<0.001
7–8 h	32.7%	30.0%	
≥9 h	8.5%	9.5%	
Kessler score >13	13.8%	12.0%	<0.001
Sex			<0.001
Male	45.1%	39.1%	
Female	54.9%	60.9%	
Average age	48.04	46.26	<0.001
Body mass index (BMI)	27.54	29.21	<0.001
Employed percentage of the population	63.1%	57.0%	<0.001

Note: only self-identified White vs. Black; does not include mixed race, Hispanic, Asian, etc.).

**Table 2 ijerph-18-01718-t002:** Bivariate correlations matrix between relevant variables of interest and covariates (*n* = 274,041).

Variable	Sleep	Emotional Distress	Physical Activity
Emotional Distress	−0.10	-	-
Physical Activity	−0.03	−0.07	-
Age	0.07	−0.04	−0.15
Sex	0.01	0.08	−0.05
Body Mass Index	−0.05	0.07	−0.07
Employment Status	0.13	0.07	−0.15

Note: All correlations were significant at *p* < 0.05.

**Table 3 ijerph-18-01718-t003:** Correlations between the relevant variables of interest and covariates for the stratified White population (*n* = 227,823).

Variable	Sleep	Emotional Distress	Physical Activity
Emotional distress	−0.08	-	-
Physical activity	−0.03	−0.08	
Age	0.07	−0.02	−0.15
Sex	0.01	0.06	−0.03
Body mass index	−0.05	0.07	−0.08
Employment status	0.13	0.12	−0.15

Note: All correlations were significant at *p* < 0.05.

**Table 4 ijerph-18-01718-t004:** Correlations between the relevant variables of interest and covariates for the stratified Black population (*n* = 46,218).

Variable	Sleep	Emotional Distress	Physical Activity
Emotional distress	−0.09	-	-
Physical activity	−0.05	−0.07	-
Age	0.06	−0.03	−0.15
Sex	0.01	0.05	−0.09
Body mass index	−0.05	0.07	−0.02
Employment status	0.12	0.16	−0.15

Note: All correlations were significant at *p* < 0.05.

**Table 5 ijerph-18-01718-t005:** The moderating effect of physical activity on the relationship between healthy sleep and emotional distress predictions among Blacks and Whites (*n* = 274,041).

Variables	Odds Ratio	95% CI—Lower	95% CI—Upper	*p*-Value
Unhealthy sleep (reference: 7–8 h)	2.549	2.458	2.644	<0.001
Physical activity	0.784	0.725	0.848	<0.001
Physical activity × sleep	0.929	0.886	0.975	0.003
Age	0.985	0.984	0.986	<0.001
Sex (reference: male)	1.316	1.284	1.348	<0.001
BMI	1.024	1.022	1.026	<0.001
Employment status (reference: employed)	2.338	2.280	2.398	<0.001

Note: Unhealthy sleep is defined as <7 or >8 h and coded dichotomously. BMI was calculated with self-reported height and weight. Physical activity was defined as whether a participant regularly performed physical activity (moderate and/or vigorous.) Emotional Distress was coded dichotomously using the Kessler 6 scale, in which a score of 14 or greater indicated Emotional Distress.

**Table 6 ijerph-18-01718-t006:** The moderating effect of physical activity on the relationship between healthy sleep and emotional distress predictions among Whites (*n* = 227,823).

Variables	Odds Ratio	95% CI—Lower	95% CI—Upper	*p*-Value
Unhealthy sleep (reference: 7–8 h)	2.603	2.499	2.712	<0.001
Physical activity	0.705	0.677	0.733	<0.001
Physical activity × sleep	0.924	0.876	0.975	0.004
Age	0.985	0.984	0.985	<0.001
Sex (reference: male)	1.343	1.308	1.380	<0.001
BMI	1.026	1.024	1.028	<0.001
Employment status (reference: employed)	2.297	2.233	2.363	<0.001

Note: Unhealthy sleep is defined as <7 or >8 h and coded dichotomously. BMI was calculated with self-reported height and weight. Physical activity was defined as whether a participant regularly performed physical activity (moderate and/or vigorous.) Emotional Distress was coded dichotomously using the Kessler 6 scale, in which a score of 14 or greater indicated Emotional Distress.

**Table 7 ijerph-18-01718-t007:** The moderating effect of physical activity on the relationship between healthy sleep and emotional distress predictions among Blacks (*n* = 46,218).

Variables	Odds Ratio	95% CI—Lower	95% CI—Upper	*p*-Value
Unhealthy sleep (reference: 7–8 h)	2.426	2.237	2.630	<0.001
Physical activity	0.847	0.776	0.925	<0.001
Physical activity × sleep	0.903	0.807	1.009	0.072
Age	0.986	0.984	0.987	<0.001
Sex (reference: male)	1.255	1.185	1.335	<0.001
BMI	1.019	1.016	1.023	<0.001
Employment status (reference: employed)	2.534	2.392	2.685	<0.001

Note: Unhealthy sleep is defined as <7 or >8 h and coded dichotomously. BMI was calculated with self-reported height and weight. Physical activity was defined as whether a participant regularly performed physical activity (moderate and/or vigorous.) Emotional Distress was coded dichotomously using the Kessler 6 scale, in which a score of 14 or greater indicated Emotional Distress.

**Table 8 ijerph-18-01718-t008:** The moderating effect of physical activity on the relationship between short sleep and emotional distress among Blacks and Whites (*n* = 274,041).

Variables	Odds Ratio	95% CI—Lower	95% CI—Upper	*p*-Value
Short sleep (reference: 7–8 h)	2.752	2.647	2.862	<0.001
Physical activity	0.731	0.705	0.758	<0.001
Physical activity × short sleep	0.933	0.887	0.982	0.007
Age	0.985	0.984	0.986	<0.001
Sex (reference: male)	1.315	1.283	1.347	<0.001
BMI	1.024	1.022	1.025	<0.001
Employment status (reference: employed)	2.423	2.362	2.486	<0.001

Note: Short sleep was defined as less than 7 h. BMI was calculated with self-reported height and weight. Physical activity was defined as whether a participant regularly performed physical activity (moderate and/or vigorous.) Emotional Distress was coded dichotomously using the Kessler 6 scale, in which a score of 14 or greater indicated Emotional Distress.

**Table 9 ijerph-18-01718-t009:** The moderating effect of physical activity on the relationship between long sleep and emotional distress among Blacks and Whites (*n* = 274,041).

Variables	Odds Ratio	95% CI—Lower	95% CI—Upper	*p*-Value
Long sleep (reference: 7–8 h)	2.037	1.926	2.154	<0.001
Physical activity	0.731	0.705	0.758	<0.001
Physical activity × long sleep	0.851	0.787	0.920	<0.001
Age	0.985	0.984	0.986	<0.001
Sex (reference: male)	1.315	1.283	1.347	<0.001
BMI	1.024	1.022	1.025	<0.001
Employment status (reference: employed)	2.423	2.362	2.486	<0.001

Note: Long sleep was defined as more than 8 h. BMI was calculated with self-reported height and weight. Physical activity was defined as whether a participant regularly performed physical activity (moderate and/or vigorous.) Emotional Distress was coded dichotomously using the Kessler 6 scale, in which a score of 14 or greater indicated Emotional Distress.

**Table 10 ijerph-18-01718-t010:** The moderating effect of physical activity on the relationship between short sleep compared to healthy sleep and emotional distress among Whites (*n* = 227,823).

Variables	Odds Ratio	95% CI—Lower	95% CI—Upper	*p*-Value
Short sleep (reference: 7–8 h)	2.809	2.688	2.935	<0.001
Physical activity	0.707	0.679	0.736	<0.001
Physical activity × short sleep	0.927	0.876	0.981	0.008
Age	0.985	0.984	0.985	<0.001
Sex (reference: male)	1.342	1.306	1.379	<0.001
BMI	1.026	1.024	1.028	<0.001
Employment status (reference: employed)	2.379	2.312	2.448	<0.001

Note: Short sleep was defined as less than 7 h. BMI was calculated with self-reported height and weight. Physical activity was defined as whether a participant regularly performed physical activity (moderate and/or vigorous.) Emotional Distress was coded dichotomously using the Kessler 6 scale, in which a score of 14 or greater indicated Emotional Distress.

**Table 11 ijerph-18-01718-t011:** The moderating effect of physical activity on the relationship between long sleep compared to healthy sleep and emotional distress predictions among Whites (*n* = 227,823).

Variables	Odds Ratio	95% CI—Lower	95% CI—Upper	*p*-Value
Long sleep (reference: 7–8 h)	2.097	1.970	2.232	<0.001
Physical activity	0.707	0.679	0.736	<0.001
Physical activity × long sleep	0.858	0.788	0.935	<0.001
Age	0.985	0.984	0.985	<0.001
Sex (reference: male)	1.342	1.306	1.379	<0.001
BMI	1.026	1.024	1.028	<0.001
Employment status (reference: employed)	2.379	2.312	2.448	<0.001

Note: Long sleep was defined as more than 8 h. BMI was calculated with self-reported height and weight. Physical activity was defined as whether a participant regularly performed physical activity (moderate and/or vigorous.) Emotional Distress was coded dichotomously using the Kessler 6 scale, in which a score of 14 or greater indicated Emotional Distress.

**Table 12 ijerph-18-01718-t012:** The moderating effect of physical activity on the relationship between short sleep and emotional distress predictions among Blacks (*n* = 46,218).

Variables.	Odds Ratio	95% CI—Lower	95% CI—Upper	*p*-Value
Short sleep (reference: 7–8 h)	2.640	2.424	2.875	<0.001
Physical activity	0.851	0.779	0.930	<0.001
Physical activity × short sleep	0.908	0.808	1.021	0.107
Age	0.986	0.984	0.987	<0.001
Sex (reference: male)	1.266	1.185	1.331	<0.001
BMI	1.019	1.015	1.023	<0.001
Employment status (reference: employed)	2.636	2.487	2.794	<0.001

Note: Short sleep was defined as less than 7 h. BMI was calculated with self-reported height and weight. Physical activity was defined as whether a participant regularly performed physical activity (moderate and/or vigorous.) Emotional Distress was coded dichotomously using the Kessler 6 scale, in which a score of 14 or greater indicated Emotional Distress.

**Table 13 ijerph-18-01718-t013:** The moderating effect of physical activity on the relationship between long sleep and emotional distress predictions among Blacks (*n* = 46,218).

Variables	Odds Ratio	95% CI—Lower	95% CI—Upper	*p*-Value
Long sleep (reference: 7–8 h)	1.848	1.630	2.095	<0.001
Physical activity	0.851	0.779	0.930	<0.001
Physical activity × long sleep	0.775	0.644	0.934	0.007
Age	0.986	0.984	0.987	<0.001
Sex (reference: male)	1.266	1.185	1.331	<0.001
BMI	1.019	1.015	1.023	<0.001
Employment status (reference: employed)	2.636	2.487	2.794	<0.001

Note: Long sleep was defined as more than 8 h. BMI was calculated with self-reported height and weight. Physical activity was defined as whether a participant regularly performed physical activity (moderate and/or vigorous.) Emotional Distress was coded dichotomously using the Kessler 6 scale, in which a score of 14 or greater indicated Emotional Distress.

## Data Availability

Data available in a publicly accessible repository.

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
