# Peer review of "The Moderating Effect of Physical Activity on the Relationship between Sleep and Emotional Distress and the Difference between Blacks and Whites: A Secondary Data Analysis Using the National Health Interview Survey from 2005–2015"

_ijerph, 2021, doi:10.3390/ijerph18041718_

Round 1

Reviewer 1 Report

The manuscript that was sent to me for review is an example of a practically model scientific publication.

The research topic is important and topical. The study was planned and performed with the utmost care. The description of the methods and results is correct. The discussion was briefly summarized, with the appropriate selection of literature. The right conclusions were drawn.

Only as a reviewer obligation, in order to make the results easier to read, add figures to the manuscript summarizing the most important observations.Only the figures summarizing the results are missing in the manuscript.

Author Response

Response:  Thank you for the feedback on this manuscript.  The figures have been modified based on other reviewers' suggestions and some are included in supplementary material.

Reviewer 2 Report

The current study examines the link among sleep, emotional distress and physical activity, and how its moderating effect differs between black and white adults. 

This is a well-designed and well-written study. The major limitations are the self-reported measures used (as mentioned in the limitations section) and the different percentage of the two ethnic groups. No further major comments. 

Author Response

Response: Thank you for your review and recommendation for publication for our manuscript.

Reviewer 3 Report

December 10, 2020

Review of the paper

The Moderating Effect of Physical Activity on the Relationship Between Sleep and Emotional Distress, and Difference Across Blacks vs Whites: A Secondary Data Analysis Using the National Health Interview Survey from 2005-2015

by Jesse Moore, Shannique Richards, Collin Popp, Laronda Hollimon, Marvin Reid, Girardin Jean-Louis, and Azizi A. Seixas

General Comments

In this paper, the Authors study the influence of physical activity on emotional distress and sleep duration. They try to answer the question if the relation above differs between Blacks and Whites. This study was based on data from the 2005-2015 National Health Interview Survey a nationally 21 representative sample of 416,152 participants. The survey treats an interesting health problem. The results are of some value.

Major Comments

I find the idea presented in the paper interesting but I have the following comments and concerns: 

1. Logistics regression is not a new concept. What is the added value of this paper? This should be clearly pointed out in the Abstract as well as in the main text.

2. How differs proposed approach from this in the papers: Seixas AA, Auguste E, Butler M, James C, Newsome V, Auguste E, da Silva Fonseca VA, Schneeberger A, Zizi F, Jean-Louis G. Differences in short and long sleep durations between blacks and whites attributed to emotional distress: analysis of the National Health Interview Survey in the United States. Sleep Health. 2017 Feb;3(1):28-34. DOI: 10.1016/j.sleh.2016.11.003. Epub 2016 Dec 13. PMID: 28346147; PMCID: PMC6911358

and

Siahpush M, Robbins RE, Ramos AK, Michaud TL, Clarke MA, King KM. Does Difference in Physical Activity Between Blacks and Whites Vary by Sex, Income, Education, and Region of Residence? Results from 2008 to 2017 National Health Interview Surveys. J Racial Ethn Health Disparities. 2019 Oct;6(5):883-891. DOI: 10.1007/s40615-019-00586-9. Epub 2019 Apr 19. PMID: 31004290.

3. Conclusions should be reformulated to stress the result obtained by the Authors.

Minor Comments

1. Lack of section Background in the Abstract. The first sentence of the Abstract is difficult to read due to the grammatical issue.

According to the journal requirement (i.e. The abstract should be a total of about 200 words maximum. The abstract should be a single paragraph and should follow the style of structured abstracts, but without headings: 1) Background: Place the question addressed in a broad context and highlight the purpose of the study; 2) Methods: Describe briefly the main methods or treatments applied. Include any relevant preregistration numbers, and species and strains of any animals used. 3) Results: Summarize the article's main findings; and 4) Conclusion: Indicate the main conclusions or interpretations. The abstract should be an objective representation of the article: it must not contain results which are not presented and substantiated in the main text and should not exaggerate the main conclusions.) The Abstract should be reformulated.

2. Throughout the manuscript, spaces are missing before the citation, and there are unnecessary spaces within the parentheses.

3. In section 2. Materials and Methods there is an unnecessary period before the section number.

4. The spelling of Black and Whites should be standardized throughout the manuscript (uppercase or lowercase).

5. English should be improved: There are some typos and grammatical writing issues, please have a very careful check of the writing. There are some repetitions.

Final Comments

The idea presented and developed in the paper seems to be promising and interesting, but due to the above concerns, I could recommend this paper for publication provided that the above comments/questions will be carefully addressed. At this moment I do not recommend the paper for publication.

Author Response

Jesse Moore

Response to Reviewer 3

Open Review

English language and style

( ) Extensive editing of English language and style required
(x) Moderate English changes required
( ) English language and style are fine/minor spell check required
( ) I don't feel qualified to judge about the English language and style

Yes

Can be improved

Must be improved

Not applicable

Does the introduction provide sufficient background and include all relevant references?

( )

(x)

( )

( )

Is the research design appropriate?

( )

( )

(x)

( )

Are the methods adequately described?

( )

( )

(x)

( )

Are the results clearly presented?

( )

(x)

( )

( )

Are the conclusions supported by the results?

( )

( )

(x)

( )

Comments and Suggestions for Authors

December 10, 2020

Review of the paper

Differences in the Moderating Effect of Physical Activity on the Relationship Between Sleep and Emotional Distress between Blacks and Whites in the United States: A Trends Analysis Using the National Health Interview Survey from 2005-2015

by Jesse Moore, Shannique Richards, Collin Popp, Laronda Hollimon, Marvin Reid, Girardin Jean-Louis, and Azizi A. Seixas

General Comments

In this paper, the Authors study the influence of physical activity on emotional distress and sleep duration. They try to answer the question if the relation above differs between Blacks and Whites. This study was based on data from the 2005-2015 National Health Interview Survey a nationally  representative sample of 416,152 participants. The survey treats an interesting health problem. The results are of some value.

Major Comments

I find the idea presented in the paper interesting but I have the following comments and concerns: 

  1. Logistics regression is not a new concept. What is the added value of this paper? This should be clearly pointed out in the Abstract as well as in the main text.

Response:  We thank the reviewer for their feedback about the need to underscore the added value of our manuscript. This comment allowed us to refine the public health and scientific importance of our study. To address this comment, we highlighted three major contributions of our manuscript.  First, our study investigated associations among sleep, physical activity and emotional distress among a racially/ethnically diverse sample of Blacks and Whites in the US using national trends data.  in the NHIS, which to our knowledge has not been done in previously. Second, we found from stratified race/ethnicity stratified analysis that the attenuating and protective effect of physical activity is different between Whites and  Blacks. Third, the implication of our findings can affect public health programs and initiatives as it appears that some racial/ethnic groups may not confer the same health benefits from physical activity as Whites. This finding also provides scientific grounding to explore in future research.  The manuscript will be edited to make this clear. (Page 10, 11 and 12.)

  1. How differs proposed approach from this in the papers: Seixas AA, Auguste E, Butler M, James C, Newsome V, Auguste E, da Silva Fonseca VA, Schneeberger A, Zizi F, Jean-Louis G. Differences in short and long sleep durations between blacks and whites attributed to emotional distress: analysis of the National Health Interview Survey in the United States. Sleep Health. 2017 Feb;3(1):28-34. DOI: 10.1016/j.sleh.2016.11.003. Epub 2016 Dec 13. PMID: 28346147; PMCID: PMC6911358

and

Siahpush M, Robbins RE, Ramos AK, Michaud TL, Clarke MA, King KM. Does Difference in Physical Activity Between Blacks and Whites Vary by Sex, Income, Education, and Region of Residence? Results from 2008 to 2017 National Health Interview Surveys. J Racial Ethn Health Disparities. 2019 Oct;6(5):883-891. DOI: 10.1007/s40615-019-00586-9. Epub 2019 Apr 19. PMID: 31004290.

Response:  Although our manuscript builds on the approaches from these papers, it examined the moderating role of physical activity on the relationship between sleep and emotional distress between Blacks and Whites.  Our paper is also different from the two mentioned papers because we treat sleep duration as a predictor and emotional distress as an outcome.  Our manuscript is supported by previous work indicating that physical activity is associated with sleep duration and stress/mental health outcomes, and emotional distress.

  1. Conclusions should be reformulated to stress the result obtained by the Authors.

Response: The conclusion section of the manuscript will be edited with this feedback. (Page 12)

Our study also demonstrates that both short and long sleep have a greater positive relationship with emotional distress than healthy sleep, but short sleep is the most positively associated. 

Minor Comments

  1. Lack of section Background in the Abstract. The first sentence of the Abstract is difficult to read due to the grammatical issue.

According to the journal requirement (i.e. The abstract should be a total of about 200 words maximum. The abstract should be a single paragraph and should follow the style of structured abstracts.

Response: The first sentence and the rest of the abstract has been reformatted (Page 1.)

1) Background: Unhealthy sleep durations (short and long sleep) are associated with Emotional distress (ED).  Minority populations, specifically Blacks, are more burdened with unhealthy sleep duration and ED.  The ameliorative effect of physical activity (PA) on ED and sleep duration may provide insight on how to reduce burden among Blacks and other minorities.  However, it is unclear whether PA attenuates the relationship between sleep and ED, and if this relationship differs by race. 2) Methods: We analyzed data from the nationally representative 2005-2015 National Health Interview Survey (NHIS) dataset. ED, physical activity, and sleep duration were collected through self-report.  Regression analyses investigated the moderating effect of PA on the relationship between sleep and ED (adjusting for age, sex, BMI, and employment status) and stratified by race.  3) Results: We found that sleep duration is independently associated with ED.  Physical activity moderated the relationship between sleep and ED, full population and Whites, not Blacks.  4) Conclusion: PA moderated the relationship between short, average, or long sleep and ED, but in stratified analyses this was only evident for Whites, suggesting Blacks receive differing protective effects of physical activity. Further research should be performed to understand the connection of physical activity to sleep and mental health.  Keywords: Emotional Distress, Mental Health, Physical Activity, Sleep, Minority Health, Race

Reviewer 4 Report

ijerph-1042161

This is a large epidemiological study on sleep and physical activity focusing on racial difference. Racial differences may be an important issue in the society, but this manuscript need improvements.

Major points:

This study is focusing on difference in Black and White. It needs reasonings why not including other races such as Hispanic/Latino and Asian. According to some report (https://www.pewresearch.org/2020/09/23/the-changing-racial-and-ethnic-composition-of-the-u-s-electorate/), Hispanics became second largest race in US. Hispanics are mainly increased in California and Nevada, but White are decreasing in Nevada with the similar ratio. Number of Asians were much less, but their increase is nationwide and looks more stable.

Please, at least, compare other races and present the results as supplementary materials.

From Tables 3 and 4, differences between black and white were found correlation employment vs. emotional distress/physical activity (Correlation between employment and emotional distress was 0.07 in White, while that was 0.15 in Black. Correlation between employment and physical activity was – 0.16 in White, while that was 0.03 in Black). However, logistic regression results for employment were not presented in tables 5, 6 and 7.

Pleas present which affects emotional distress more, shorter sleep duration or longer sleep duration? This may be an interesting point from your data.

Other confounding factors (such as employment, social status, education) might affect emotional distress. In lines 164-165, authors present “potentially relevant covariates that have been identified in past research as affecting physical activity and sleep (age, sex, employment status, race and BMI)” without references. Confounding factors are important and may need more descriptions in introduction or discussion.

Author Response

Jesse Moore - Response

Reviewer 4

Open Review

English language and style

( ) Extensive editing of English language and style required
( ) Moderate English changes required
( ) English language and style are fine/minor spell check required
(x) I don't feel qualified to judge about the English language and style

Yes

Can be improved

Must be improved

Not applicable

Does the introduction provide sufficient background and include all relevant references?

( )

( )

(x)

( )

Is the research design appropriate?

( )

( )

(x)

( )

Are the methods adequately described?

( )

( )

(x)

( )

Are the results clearly presented?

( )

(x)

( )

( )

Are the conclusions supported by the results?

( )

(x)

( )

( )

Comments and Suggestions for Authors

ijerph-1042161

This is a large epidemiological study on sleep and physical activity focusing on racial difference. Racial differences may be an important issue in the society, but this manuscript need improvements.

Major points:

This study is focusing on difference in Black and White. It needs reasonings why not including other races such as Hispanic/Latino and Asian. According to some report (https://www.pewresearch.org/2020/09/23/the-changing-racial-and-ethnic-composition-of-the-u-s-electorate/), Hispanics became second largest race in US. Hispanics are mainly increased in California and Nevada, but White are decreasing in Nevada with the similar ratio. Number of Asians were much less, but their increase is nationwide and looks more stable.

Please, at least, compare other races and present the results as supplementary materials.

Response:  We agree with the reviewers point that Hispanics are a growing racial/ethnic group in the US. Although Hispanics and Asians make up a significant portion of the US population, the prevalence and burden of poor sleep and emotional distress is lower compared to Blacks.  Therefore, our focus on Blacks and Whites builds on gaps in previous literature that Blacks disproportionately are burdened by poor sleep and emotional distress.1 We acknowledge the importance of and need to investigating other racial/ethnic groups but feel that those analyses warrant a separate analysis and manuscript to fully capture the nuances across other racial/ethnic groups.

Pleas present which affects emotional distress more, shorter sleep duration or longer sleep duration? This may be an interesting point from your data.

 Response:  This is an interesting point, and this analysis was conducted and added to the manuscript.  There is a limited amount of literature comparing long vs. short sleep, but we found that short sleep was more strongly positively associated with emotional distress (page 10.)  This was true even with stratification across races. 

Other confounding factors (such as employment, social status, education) might affect emotional distress. In lines 164-165, authors present “potentially relevant covariates that have been identified in past research as affecting physical activity and sleep (age, sex, employment status, race and BI)” without references. Confounding factors are important and may need more descriptions in introduction or discussion.

Response:  Analyses were re-done in the manuscript to include occupation or employment as a covariate.  Occupation and employment are important factors to consider when assessing sleep and mood outcomes between Blacks and Whites, since Blacks are more likely to take jobs that require shift work or irregular schedules.2  Literature finds that age, sex, and BMI are important underlying factors of sleep and emotional distress, so they were included as covariates.3 (Page 4.)

(1)      Williams, D. R.; Yu, Y.; Jackson, J. S.; Anderson, N. B. Racial Differences in Physical and Mental Health. Socio-Economic Status, Stress and Discrimination. J. Health Psychol. 1997. https://doi.org/10.1177/135910539700200305.

(2)      Jackson, C. L.; Redline, S.; Kawachi, I.; Williams, M. A.; Hu, F. B. Racial Disparities in Short Sleep Duration by Occupation and Industry. Am. J. Epidemiol. 2013. https://doi.org/10.1093/aje/kwt159.

(3)      Williams, N. J.; Grandner, M. A.; Wallace, D. M.; Cuffee, Y.; Airhihenbuwa, C.; Okuyemi, K.; Ogedegbe, G.; Jean-Louis, G. Social and Behavioral Predictors of Insufficient Sleep among African Americans and Caucasians. Sleep Med. 2016, 18, 103–107. https://doi.org/10.1016/j.sleep.2015.02.533.

Round 2

Reviewer 3 Report

The Authors address my comments.  

Reviewer 4 Report

Authors responded well to the comments.